# Repetition of the Exhaustive Wrestling-Specific Test Leads to More Effective Differentiation between Quality Categories of Youth Wrestlers

Kreso Skugor [1], Hrvoje Karnincic [1,*], Nenad Zugaj [2], Valdemar Stajer [3] and Barbara Gilic [1,4]

[1] Faculty of Kinesiology, University of Split, 21000 Split, Croatia; kresoskugor95@gmail.com (K.S.); barbara.gilic@kifst.eu (B.G.)
[2] Faculty of Kinesiology, University of Zagreb, 10000 Zagreb, Croatia; nenadzugaj@gmail.com
[3] Faculty of Sport and Physical Education, University of Novi Sad, 21000 Novi Sad, Serbia; stajervaldemar@yahoo.com
[4] High Performance Sport Center, Croatian Olympic Committee, 10000 Zagreb, Croatia
[*] Correspondence: hrvoje.karnincic@kifst.eu

**Abstract:** This study aimed to investigate whether wrestlers of different competitive qualities (i.e., medalists vs. non-medallists) would differ in terms of specific test performance and cardiac and metabolic responses after a demanding testing protocol. The research included 29 wrestlers aged $17.62 \pm 1.86$ years divided into two performance categories: successful (medallists at the National Championships; n = 13) and less successful (non-medallists; n = 16). The variables included anthropometric indices and specific wrestling fitness test (SWFT) parameters, including the number of throws, heart rate, lactate concentration and calculated cardiac and metabolic indexes. To show differences between quality categories, Student's *t*-test and receiver operating characteristic curves (ROC) were calculated. Two-way ANOVA for repeated measurements was used to evaluate the differences in performance, cardiac, and metabolic characteristics between the test trials and quality categories. Wrestlers differed in the total number of throws ($p < 0.01$, AUC = 0.82), cardiac indices ($p < 0.03$, AUC = 0.73), and metabolic indices ($p < 0.04$, AUC = 0.75) after the second SWFT trial, with successful wrestlers reaching better results. There were no differences in the first testing trial. The findings of this study indicate that wrestlers exhibit differences in specific performance variables after undergoing an exhaustive testing protocol. Therefore, this study suggests that future research on sport-specific performance in wrestlers should include exhaustive exercise or testing protocols.

**Keywords:** physical performance; sport specific; testing; youth athletes; anaerobic capacities; combat

## 1. Introduction

Wrestling is an Olympic sport characterized by intermittent activity requiring developed technical and tactical abilities and physical capacities. Olympic-style wrestling includes Greco-Roman and Freestyle wrestling, with the match consisting of two rounds lasting 3 min with a 30 s break between the rounds [1]. Wrestling matches are extremely dynamic and include sudden attacks and counterattacks interspersed with actions for controlling the opponent [2]. Thus, wrestling matches demand constant engagement, which means that aerobic and anaerobic metabolic energy systems are involved during the match [2,3]. Specifically, the aerobic metabolic energy system provides the energy for enduring continuous effort. It enhances recovery, while the anaerobic metabolic energy system is engaged during movements under submaximal and maximal loads [3–5]. Notably, 90% of the total energy generated during a wrestling match comes from the anaerobic energy system, meaning that anaerobic glycolysis is most prevalent during the match [6,7]. Moreover, the anaerobic system produces energy for quick bursts of maximum power,

such as sudden lifts and throws, which are crucial parts of the wrestling match [8]. Therefore, as crucial wrestling movements occur in the anaerobic metabolic pathway, wrestling performance in the anaerobic energetic system has been the focus of diagnostic work [9].

One of the recent sport-specific tests that was created to mimic the physical and metabolic loads of the wrestling match is the Specific Wrestling Fitness Test (SWFT). The SWFT simulates the time and load structure of the match. Specifically, SWFT is intermittent and consists of three rounds of 30 s maximum dummy throws with 20 s rest between the rounds [10]. Indeed, it has been reported that SWFT evoked cardiac and metabolic responses similar to ones after a wrestling match (i.e., maximal heart rate and blood lactate concentration of 10–20 mmol/L) and are associated with submaximal aerobic performance variables [11,12]. However, it predominantly evaluates the anaerobic glycolytic system, in which most of the actions are performed during the wrestling match [13]. Moreover, the SWFT is valid for differentiating successful from less successful adult wrestlers in Serbia and youth wrestlers in Croatia, which implies that the SWFT can be used for evaluating the specific physical performance of wrestlers [13–15].

A wrestling match can elevate blood lactate to high concentrations of nearly 20 mmol/L, which directly influences muscle contractile mechanisms by interrupting actin-myosin cross-bridge interactions [16]. Also, the wrestling match places high demands on the heart and can increase the heart rate to the maximum value [17]. Additionally, the high neuromuscular demand of the upper and lower limbs during a wrestling match may cause decrements in performance [18]. Indeed, a study on collegiate wrestlers reported a reduction in physical performance as a result of tournament wrestling [11]. Specifically, as a wrestling match includes isometric grasping for maintaining one's position, the handgrip strength was reduced in response to tournament wrestling. Furthermore, as the majority of actions are performed with powerful leg movements, the lower body power was reduced after one match [11]. Notably, a study on top-level junior and under-23 wrestlers indicated that strength and power tests performed after exhaustive exercise are better at discriminating between elite and top-elite wrestlers [19].

The capacity to endure as long as possible and maintain the highest level of muscular strength and power is crucial for winning in combat. This is especially important in combat sports, as strength–power interactions performed despite fatigue represent a determinative factor [18,20,21]. Thus, testing physical performance after exhaustive exercise or repeated tests should be more sensitive than testing at rest for differentiating successful from less successful athletes [19]. Therefore, this study aimed to investigate whether youth wrestlers of different competitive qualities (i.e., medallists vs. non medallists) would differ in the performance, cardiac, and metabolic parameters of SWFT after repeating tests, which created a physically demanding testing protocol that mimics real-life match situations. We hypothesized that successful wrestlers would have more favourable cardiac, metabolic, and performance responses after repeated tests than would less successful wrestlers.

## 2. Materials and Methods

### 2.1. Participants

The research included 29 Greco-Roman wrestlers from Croatia aged 17 ± 1 years with training experience of 6 ± 3 and competing experience of 6 ± 3 years who were competing in the cadet and junior categories. Wrestlers were divided into two performance categories: successful wrestlers were medallists, i.e., wrestlers who won a medal at the National Championship in 2022 (n = 13), and less successful wrestlers, non-medallists (n = 16). The main characteristics of the successful wrestlers were as follows: body height, 180 ± 7 cm; body mass, 83 ± 16 kg; body mass index, 25 ± 3; body fat percentage, 15 ± 5%; and competing experience, 7 ± 4 years. The main characteristics of the less successful wrestlers were as follows: body height, 177 ± 7 cm; body mass, 76 ± 14 kg; body mass index, 24 ± 3; body fat percentage, 17 ± 6%; and competing experience, 5 ± 2. years. The inclusion criteria were at least three years of wrestling experience and participation in the National Championships. This way, researchers wanted to be sure that wrestlers have

appropriate knowledge of wrestling techniques and can execute the included tests correctly. The exclusion criterion was having any illness or medical condition that prevents wrestlers from maximally executing the tests included in the testing procedure. It is important to note that all the included wrestlers were competitors and not recreative individuals, which means that they all had highly developed physical capacities and wrestling skills. The sample size was computed through the statistical programme G*power using the data of the previous similar research with an effect size of 1.77 in the SWFT total number of throws variable, with the power of 0.95, resulting in a calculated 20 participants [22,23].

Participants were informed about the testing procedures and aims of the investigation and signed an informed consent (legal guardians signed an informed consent for participants under 18 years of age). The Ethical Board of the Faculty of Kinesiology, University of Split, approved this study (Ref. no. 2181-205-02-05-22-0012).

*2.2. Variables*

This research included anthropometric indices and specific wrestling fitness test parameters (heart rate, blood lactate, and performance indicators).

The SWFT is a relatively new specific wrestling test. The SWFT consists of 3 rounds of 30 s of maximal dummy throws followed by 20 s of rest after each throwing round (please see Figure 1 for details). Since the athlete's body mass plays a role in physical performance indicators, wrestlers perform the SWFT with a specific weight. Precisely, the weight of the dummy was allocated to each wrestler according to weight category as follows: the 55–67 kg category was tested with a 23 kg dummy, the 72–87 kg category was tested with a 25 kg dummy, and wrestlers weighing more than 90 kg were tested with a 30 kg dummy. The main result of the test was the total number of throws generated during all three throwing rounds [10].

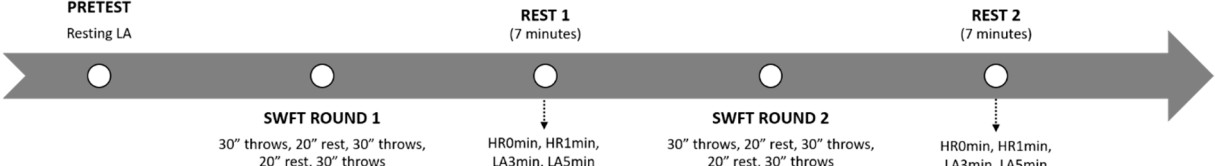

**Figure 1.** Testing procedure. Note: LA—blood lactate concentration, SWFT—Specific wrestling fitness test, HR0—heart rate directly after the test, HR1—heart rate after the first minute of recovery, LA3min—lactate concentration in the third minute of recovery, and LA5min—lactate concentration in the fifth minute of recovery.

In addition to the total number of throws, metabolic and cardiac indicators were included to provide insight into the physiological strain of the test. Specifically, the metabolic indicator was the lactate concentration in the capillary blood in the third (LA3min) and fifth minute (LA5min) of recovery, which represents achieved metabolic acidosis. The lactate concentration was measured by an experienced researcher using a portable lactate analyzer (Lactate Plus-NOVA Biomedical, MA, USA), and the results are expressed in mmol/L. All lactate samples were drawn from capillary blood drawn from the fingertip and from different fingers each time. The cardiac indicator was the frequency of the pulse, which represents the load on the cardiovascular system. Participants wore heart rate belts (POLAR H10, Polar, Inc., Lake Success, NY, USA) around the chest during the entire test. The heart rate was observed immediately after the test (HR0min) and after the first minute of recovery (HR1min), expressed in beats per minute.

The absolute performance measures (number of throws), metabolic (blood lactate concentrations), and cardiac (heart rate) response measures were integrated into performance-specific index measures of the SWFT. Specifically, the SWFT index 1 represents the cardiac response and was calculated in the same way as the judo fitness index: SWFT index 1 = (HR0 + HR1)/total number of throws [24]. The SWFT index 2 represents the cardiometabolic

response and was developed by Markovic et al. (2017), calculated as follows: SWFT index 2 = ((HR0min + HR1min)/(LA3min + LA5min)) × a total number of throws [10].

### 2.3. Testing Procedures

All testing procedures were conducted during the morning to avoid diurnal variations. First, body mass, body height, body fat percentage (skinfolds), and resting blood lactate concentrations were measured. After that, all the athletes underwent the same warm-up routine. Specifically, the warm-up consisted of 15 min of general warm-up, which included mobility exercises, running, and skipping for elevating heartbeats, and dynamic stretching exercises. Afterward, all participants practised throwing the dummy for 10 min of intermittent throwing, emphasizing the suplex throwing technique. Notably, we included experienced wrestlers who were proficient at throwing the dummy with the suplex technique.

After the warm-up, the first round of the SWFT was performed. Immediately after the first trial of SWFT, participants were resting for 7 min, and blood lactate concentrations were analyzed during the third and fifth minutes of rest. After the rest, participants again performed SWFT (the second round), after which they rested for 7 min, and all the testing procedures were the same as those used during the first resting phase. Typically, a 20 min rest is provided for wrestling/combat tests and competitions because this rest mimics the actual situation during the competition (i.e., a minimum of 20 min between two matches) [25]. However, the SWFT lasts for 3 min, which is one period of real wrestling match duration (a wrestling match consists of two rounds of minutes of fight); this is why we decided that wrestlers would have 7 min of rest, which corresponds to a work-to-rest ratio of 1:2.5. Additionally, we wanted to create a situation where the recovery ability of wrestlers would be evident, as differentiating the quality of wrestlers was the main aim of this investigation. The detailed testing procedure is shown in Figure 1.

### 2.4. Statistical Analysis

The Shapiro–Wilk W-test was used to determine the normality of the distributions of all the variables included. The means and standard deviations were included as the descriptive statistics.

Since all variables were normally distributed, the parametric tests were used to answer the research questions. To show differences between quality categories (i.e., successful vs. less successful wrestlers), the Student's *t*-test for independent sample analysis was used. Furthermore, to establish which variables are better at classifying wrestlers as successful or less successful, a receiver operating characteristic (ROC) curve was generated, with an area under the curve (AUC) greater than 0.70 indicating differences in the selected variables [26]. Also, the ROC cut-point values of the SWFT parameters were calculated [27]. Furthermore, two-way analysis of variance (ANOVA) for repeated measurements (trials × group) was used to examine the variations in performance, cardiac, and metabolic characteristics between testing trials based on quality categories.

The statistical package Statistica ver. 14 (Tibco, Palo Alto, CA, USA) was utilized for all analyses.

## 3. Results

Descriptive statistics and differences between quality categories are presented in Table 1.

According to the results of the independent sample *t*-test, wrestlers differed in terms of the SWFT performance variables and cardiac and metabolic indices during the second testing trial (i.e., SWFT2 total throws, SWFT2 INDEX HR, and SWFT2 INDEX LA), while such differences were not observed in the first testing trial. These differences were additionally confirmed with ROC (area under the curve values), with SWFT2 total throws, SWFT2 INDEX HR, and SWFT2 INDEX LA reaching values greater than 0.70 (Table 1).

Figure 2 presents the ROC and AUC of the SWFT variables. It is evident that the SWFT total number of throws during the second testing trial had the greatest sensitivity for categorizing wrestlers as successful. Moreover, the ROC identified the cut points for the observed variables with an area under the curve (AUC) greater than 0.70. Specifically, the cut point value of 24.50 for SWFT2 total throws and, value of 261.26 for SWFT2 INDEX LA were identified, meaning that wrestlers who reach scores higher than those cut points are more likely to be categorized into a better performance quality group. Moreover, wrestlers who reach a value lower than 15.16 for SWFT2 INDEX HR are more likely to be categorized into better performance quality.

**Table 1.** Descriptive statistics and differences between quality categories.

| Variables | Successful (n = 13) | | Less Successful (n = 16) | | *t*-Test | | ROC | |
|---|---|---|---|---|---|---|---|---|
| | Mean | SD | Mean | SD | *t*-Value | *p* | AUC | 95% CI |
| SWFT1 TT | 26.08 | 3.99 | 23.87 | 2.42 | 1.79 | 0.09 | 0.63 | 0.38–0.89 |
| SWFT1 INDEX HR | 13.49 | 2.29 | 14.55 | 1.58 | −1.42 | 0.17 | 0.64 | 0.42–0.86 |
| SWFT1 INDEX LA | 326.52 | 60.46 | 293.59 | 45.92 | 1.45 | 0.16 | 0.68 | 0.44–0.93 |
| SWFT2 TT | 24.5 | 2.43 | 21.6 | 2.47 | 3.05 | 0.01 | 0.82 | 0.63–1.00 |
| SWFT2 INDEX HR | 14.39 | 1.68 | 16.29 | 2.5 | −2.25 | 0.03 | 0.73 | 0.53–0.92 |
| SWFT2 INDEX LA | 287.92 | 36.18 | 254.3 | 36.22 | 2.17 | 0.04 | 0.75 | 0.54–0.96 |

Note: SWFT—Specific wrestling fitness test, TT—total throws, SWFT1—The first trial of the Specific wrestling fitness test, SWFT2—The second trial of the Specific wrestling fitness test, INDEX 1—index calculated regarding heart rate and the total number of throws, INDEX 2—index calculated regarding heart rate, lactate concentration and total number of throws, HR—Heart rate, LA—blood lactate concentration, ROC—Receiver operating characteristics, AUC—Area under the curve, and CI—confidence interval.

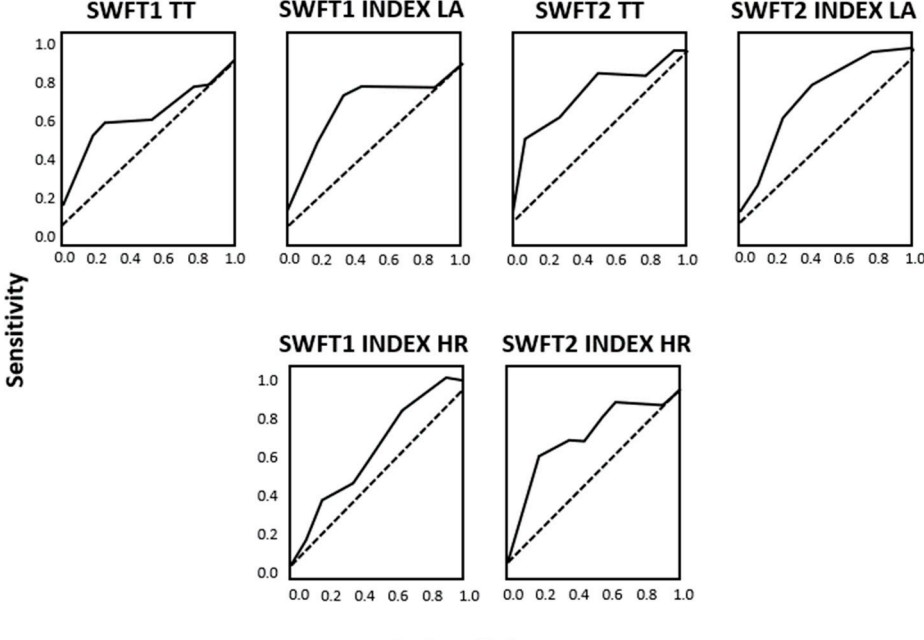

**Figure 2.** Receiver operating characteristics (ROC) curves for SWFT variables for differentiating quality categories. Note: SWFT1—The first trial of the specific wrestling fitness test, SWFT2—The second trial of the Specific wrestling fitness test, TT—total number of throws, LA—blood lactate, HR—heart rate, INDEX HR—index calculated regarding heart rate and a total number of throws, and INDEX LA—index calculated regarding heart rate, lactate concentration and a total number of throws. The dashed line represents reference line (AUC = 0.5).

Furthermore, to gain a detailed insight at which point of the testing procedure and in which parameters appear the most significant differences between performance categories,

Figure 3 is presented. Figure 3 shows a graphical representation of the two-way ANOVA results for the SWFT parameters. As shown in Figure 3A, the most visible changes between quality categories occurred at the first throwing round during the second SWFT trial. Lactate concentrations (Figure 3B) and heart rate responses (Figure 3C) were similar among the quality categories across the testing procedure.

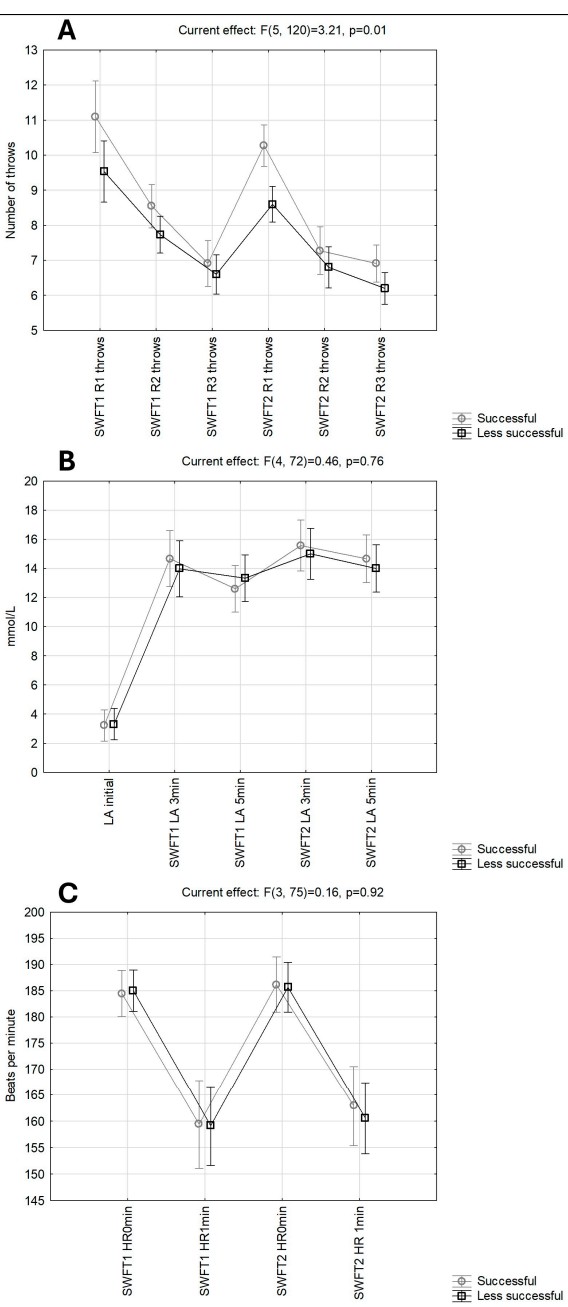

**Figure 3.** Two-way ANOVA for the SWFT throws cardiac and metabolic variables: (**A**) SWFT number of throws; (**B**) lactate concentrations; (**C**) heart rate. Note: SWFT1—The first trial of the specific wrestling fitness test, SWFT2—The second trial of the Specific wrestling fitness test, LA—blood lactate concentration, HR0—heart rate directly after the test, HR1—heart rate after the first minute of recovery, LA3min—lactate concentration in the third minute of recovery, and LA5min—lactate concentration in the fifth minute of recovery.

## 4. Discussion

The main objective of this study was to investigate whether wrestlers of different competitive qualities would differ in the performance, cardiac, and metabolic parameters in SWFT after repeating tests which created physically demanding testing protocols. According to the objectives, the main findings of this research are as follows: (i) Wrestlers of different competitive qualities did not differ in cardiac or metabolic variables in either of the SWFT trials; (ii) Wrestlers of different competitive qualities differed in terms of performance variables; successful wrestlers achieved a greater number of SWFT throws and had more favourable SWFT indexes in the second testing trial than did less successful wrestlers. Therefore, our study hypothesis can be partially accepted.

### 4.1. Differences in Cardiac and Metabolic Variables during Repeated Trials of Specific Wrestling Fitness Tests

The finding that cardiac and metabolic variables did not differ between wrestlers of different competitive qualities in either testing trial can be explained as follows. The noted results are supported by the results of previously published studies using a similar methodology. Specifically, a study on adult wrestlers from Serbia reported that the cardiac and metabolic responses of national, first, and second-league wrestlers did not differ according to the SWFT [13]. Moreover, a study on Croatian wrestlers revealed no significant differences in the La concentrations accumulated during a wrestling match between the national team and club-level wrestlers [7]. This could be explained by the fact that the SWFT and the biggest part of the wrestling match are predominantly anaerobic-glycolytic activities, which lead to high production and accumulation of blood La, placing high demands on aerobic metabolism (cardiovascular functions) to remove La from the blood during recovery [7,13]. Therefore, wrestling-specific performance corresponds to match requirements and leads to adaptations in cardiac and metabolic outputs [22].

To simplify, more successful wrestlers had advanced cardiac and metabolic adaptations, which allowed them to achieve better results and to resist fatigue during the testing trials, even though their absolute measures of physiological response were similar to those of less successful wrestlers. A similar explanation was provided by Marković, Toskić, Kukić, Zarić, and Dopsaj [13], who noted that wrestlers at higher levels could deliver larger amounts of oxygen to muscles included in the test for the same HR and execute a greater number of throws for the same La concentrations, which was explained by the increased oxygen capacity and higher contraction intensity of the muscle resulting from peripheral adaptation [13,28]. A study on Croatian wrestlers aged 15–20 years that examined the dynamics of blood lactate production hypothesized that training-induced metabolic adaptations enable high-quality wrestlers to operate for longer periods and under greater loads, and they also create more blood lactate at greater intensities and remove it faster [29]. This might also be affected by increased buffering capacity, which enables well-trained athletes to generate ATP via glycolysis [30].

According to the results of our study and those of similar previous studies, the absolute values of cardiac and metabolic variables (i.e., HR and La) are not accurate representations of physiological response because they do not consider the efficacy of the work that the wrestler is performing [31]. Therefore, cardiac and metabolic responses to effort should be adjusted to real performance because wrestlers with greater performance and the same cardiac and metabolic responses have better specific training adaptations and overall physical performance. Thus, the creators of the SWFT have proposed using index values that take into account cardiac and metabolic responses calculated together with performance values (i.e., number of throws) [10]. Indeed, our study revealed that successful wrestlers had favourable SWFT index HR and index LA during the second test round of the SWFT, which represent cardiac and metabolic responses that accounted for the total number of throws performed. However, it must be emphasized that these differences were noted only in the second test round, which is the main novelty of this study and is described in the following paragraph.

*4.2. Differences in Performance on Repeated Trials of Specific Wrestling Fitness Tests*

The result that successful wrestlers performed more SWFT total throws and had better SWFT index HR and LA in the second testing trial than did less successful wrestlers is the most interesting result of this study. Indeed, this result supports the hypothesis that wrestlers of better performance quality can better endure the physical load which supports the statement that strength–power interactions performed despite fatigue represent a determinative factor in combat [32]. Also, the result that the second testing round was better for discriminating performance quality supported the results of a previous study which concluded that strength and power tests performed after exhaustive exercise are better at discriminating between elite and top-elite wrestlers [19]. Our results can be compared to those of previous studies using the same test (i.e., the SWFT) for differentiating wrestlers according to performance and competitive quality. Specifically, a study on Serbian wrestlers evaluated the differences in performance among the national team, first-league, and second-league wrestlers and reported that higher-level wrestlers outperformed lower-level wrestlers in terms of the number of throws and the SWFT cardiac and metabolic indices [13]. However, we tested wrestlers on repeated trials of SWFTs and revealed that during the second testing trial, the most significant difference was observed between the performance qualities of the wrestlers.

The most decisive moments of wrestling combat appear when wrestlers are exhausted, tired, or unable to respond quickly to their opponent's actions [11]. Therefore, the finding that the second round of the SWFT was more sensitive than the first round of the SWFT supports this previous notion: successful wrestlers managed to perform a greater number of throws when fatigued. Additionally, it is important to mention from the results of this investigation that the most significant differences in SWFT performance appeared in the first round of throws during the second SWFT trial (please see results for details), which also indicates that more successful wrestlers were able to recover better and faster from the first SWFT trial. Notably, a study that investigated the acute physiological changes caused by wrestling tournaments reported that muscle damage markers (i.e., lactate dehydrogenase, creatine kinase, and interleukin) significantly increased during a one-day tournament, which indicates that wrestling matches place significant physiological demands on athletes and affect their performance [33]. Additionally, a study which evaluated psychological and performance changes during a one-day wrestling tournament reported a progressive rise in fatigue rating, muscle damage markers, and the inflammatory response which supported that performance is affected especially during the later rounds of the tournament [34]. Thus, better muscular capacity and physical performance of the wrestler can enable them to endure high physiological demands and stimulate recovery between matches or, in this case, between testing trials.

*4.3. Limitations and Strengths*

The main limitation of this study is that wrestlers were tested at the end of the competition season, and testing during different points of the season could lead to different results. Thus, a similar testing procedure should be conducted at several main points during the competition season (e.g., beginning, mid-season, and end of the season among the same wrestlers). Additionally, we included only male participants, so the results and conclusions from this study cannot be generalized to female participants. Moreover, the relatively small sample size does not warrant generalizations of the results, which means that similar future studies should try to include a larger number of athletes. Also, more direct parameters such as muscle oxygenation, maximal oxygen uptake, mitochondrial respiration, liver function test for determining the lactate clearance, and pulmonary function test, which could improve the understanding of the observed issue, were not conducted in this research, but are suggested to be evaluated in future studies.

The main strength of this study is that cardiac, metabolic, and performance variables were recorded at several time points during the testing procedure, which helps to track the response of the athletes to the given effort. In this way, it was possible to determine

in detail which parameters are decisive for optimal performance and which are the most important parameters for competitive success. Also, the strength of this research is that competitive-level wrestlers were included, as it is always challenging for researchers to include elite athletes.

### *4.4. Practical Implications*

The development of tests and testing procedures that can predict wrestling performance has great usefulness for coaches and competitors. Regular monitoring of sport-specific performance is important for increasing the odds of success in competitions. The result that the competitive quality of the wrestlers was distinguished after the second testing trial implies that tests should be performed after a specific exhaustive exercise protocol and not at full rest to be more effective at determining the performance quality of wrestlers. In other words, the quality of the wrestler is determined during the decisive moments of the fight, which occur when the wrestlers are usually tired and exhausted, meaning that the one who can endure and overcome fatigue will be able to conduct actions that determine the match outcome (i.e., throwing the opponent). Thus, the practical implication of this study is that coaches and sports scientists should be advised to include more sport-specific tests and to conduct them after a standardized exhaustive protocol.

Moreover, the finding that cardiac and metabolic responses (HR and LA concentrations) did not distinguish the quality of the wrestlers, while the performance and index variables did, could lead to the suggestion that SWFT can be used with and without measuring HR and LA concentrations. This is important to note because heart rate and lactate monitoring are expensive and usually unavailable for coaches, while performance variables (i.e., number of throws) during SWFTs are relatively easy for coaches to measure.

### 5. Conclusions

Wrestlers differed according to competitive quality in terms of the specific performance variables (i.e., SWFT total throws) after the exhaustive testing protocol, between the two testing trials. The findings of this study indicate that wrestlers exhibit differences in specific performance variables after undergoing an exhaustive testing protocol, and that wrestlers at higher levels could deliver larger amounts of oxygen to muscles included in the test for the same HR and execute a greater number of throws for the same La concentrations. This finding suggested that the performance of wrestlers in sport-specific actions is better assessed when they are tired, as the exhaustive protocol may reveal their true capabilities and distinguish between different performance levels. These findings have implications for the evaluation and training of wrestlers, highlighting the importance of incorporating sport-specific tests after exhaustive exercise, match or testing protocols to accurately assess their performance and tailor training programs accordingly. Therefore, future research on sport-specific performance in wrestlers in all age groups and weight categories should be conducted after exhaustive exercise or testing protocols.

**Author Contributions:** Conceptualization, V.S. and H.K.; methodology, B.G.; investigation, N.Z. and K.S.; resources, N.Z.; data curation, V.S.; writing—original draft preparation, K.S. and B.G.; writing—review and editing, V.S. and H.K.; visualization, V.S.; supervision, H.K.; funding acquisition, H.K. All authors have read and agreed to the published version of the manuscript.

**Funding:** This research received no external funding.

**Institutional Review Board Statement:** The study was conducted in accordance with the Declaration of Helsinki and approved by the Institutional Review Board Faculty of Kinesiology, University of Split (Ref. no. 2181-205-02-05-22-0012).

**Informed Consent Statement:** Informed consent was obtained from all subjects involved in the study.

**Data Availability Statement:** Data are available upon reasonable request.

**Acknowledgments:** The authors are grateful to the athletes and coaches who helped during the testing procedures.

**Conflicts of Interest:** The authors declare no conflicts of interest.

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
