# Peer review of "Repetition of the Exhaustive Wrestling-Specific Test Leads to More Effective Differentiation between Quality Categories of Youth Wrestlers"

_applsci, doi:10.3390/app14093677_

Round 1
Reviewer 1 Report
Comments and Suggestions for Authors
This study examined whether wrestlers of different competitive levels (medalists vs. non-medalists) differed in fitness test performance, cardiac, and metabolic responses following a rigorous testing protocol. The research involved 29 wrestlers aged 15-17.62 years, categorized based on performance. Results indicated variations in specific performance variables, suggesting the importance of including exhaustive testing protocols in future research on wrestler performance.
COMMENTS:
1. In a previous study, this group had found in advanced greco-roman wrestlers that generic fitness tests show no correlation with SWFT, there were no differences between wrestlers based on performance categories, and wrestlers showed no variations based on weight categories.
The current study continues on a similar line.
2. Their results corroborate an already published 2022 Serbian study which they cited "Sensitivity of Field Tests for Assessment of Wrestlers Specific Fitness". The article reads very similar to this study and appears to lack novelty.
3. This study could be made better by including other cardiometabolic parameters of wrestlers like insulin sensitivity, blood pressure, VO2max etc.
4. I wonder if the authors can comment on the mitochondrial respiration that might account for differences. Would it be possible to assess using seahorse assay?
Also, mitochondrial fragmentation should be assessed to understand biochemical differences between wrestlers and build on the previous Serbian study.
5. The wrestlers could be given a liver function test to see whether the wrestlers differ in their lactate clearance.
6. A pulmonary function test to assess whether the wrestlers vary in their breathing.
Comments on the Quality of English Language
English only needs minor revision.
Author Response
Reviewer 1
This study examined whether wrestlers of different competitive levels (medalists vs. non-medalists) differed in fitness test performance, cardiac, and metabolic responses following a rigorous testing protocol. The research involved 29 wrestlers aged 15-17.62 years, categorized based on performance. Results indicated variations in specific performance variables, suggesting the importance of including exhaustive testing protocols in future research on wrestler performance.
COMMENTS:
- In a previous study, this group had found in advanced greco-roman wrestlers that generic fitness tests show no correlation with SWFT, there were no differences between wrestlers based on performance categories, and wrestlers showed no variations based on weight categories.
The current study continues on a similar line.
RESPONSE: Yes, this research is the continuation of that previously published research. The current study tried to answer the specific research question which has been elucidated in the first study published in 2023. Precisely, we concluded that wrestlers did not differ in SWFT according to performance category but we wanted to investigate whether the difference would appear after creating the more physically demanding testing protocol as it is known that the capacity to endure more and to recover faster between the rounds is extremely important for winning the combat (i.e., being categorised as more successful athlete).
- Their results corroborate an already published 2022 Serbian study which they cited "Sensitivity of Field Tests for Assessment of Wrestlers Specific Fitness". The article reads very similar to this study and appears to lack novelty.
RESPONSE: We did indeed follow a similar line of research, however, we aimed to investigate whether repeating the test would elucidate the differences between the categories. Moreover, we tracked wrestlers' metabolic, cardiac and performance indicators throughout the testing protocol (Please see Figure 1) which enabled us to see clearer and more detailed responses throughout the protocol. This way, it was possible to see whether more successful wrestlers could recover faster, better endure more physical load and have better overall performance. Also, we believe that study can have important practical applications as it is suggested how physical performance testing should be conducted by coaches. We believe that this is the main novelty of the study. However, throughout the study procedure, we identified several parameters which should be addressed in future studies (i.e., tracking VO2max, muscle oxygenation parameters, hemodynamics etc.) and we would definitely try to conduct it in future.
- This study could be made better by including other cardiometabolic parameters of wrestlers like insulin sensitivity, blood pressure, VO2max etc.
RESPONSE: Indeed, this adds to the previous comment. The authors of the current study planned to include muscle oxygenation (using portable NIRS sensors) placed on specific muscles which are biomechanically the most important agonists and synergists of the movement required in SWFT (i.e., m. biceps femoris, m. erector spinae, m. rectus femoris, mm. intercostalis). The authors did a pilot investigation which did not appear to be good as wrestling places extreme forces on the body and on the dummy which resulted in an inability to safely attach the sensors to the specific muscles without hampering the hemodynamics by sticking the tape too hard. However, we are currently discussing this with the experts in that field and hope to solve this issue. Nevertheless, we cannot include in the current study something that hasn’t been done but we will definitely consider it for future research. Thank you!
- I wonder if the authors can comment on the mitochondrial respiration that might account for differences. Would it be possible to assess using seahorse assay?
Also, mitochondrial fragmentation should be assessed to understand biochemical differences between wrestlers and build on the previous Serbian study.
- The wrestlers could be given a liver function test to see whether the wrestlers differ in their lactate clearance.
- A pulmonary function test to assess whether the wrestlers vary in their breathing.
RESPONSES: Thank you for these valuable comments. This study included cardiac (heart rate) and metabolic (lactate concentrations) responses to the testing protocol. It would be better for sure to include other more detailed parameters as you suggested, however, we do not have the sophisticated equipment, laboratory and adequate funding so we tried to answer the initial research question by the testing procedures described in the manuscript. If you have a suggestion on what laboratory or researchers could help with conducting this, we will be happy to contact them and suggest collaboration! Nevertheless, we are unable to do any of the tests at the moment but we tried to improve the manuscript by explaining in more detail what is the novelty of this study and what could be done in future ones.
We included your suggestions in the Limitations sections, for other researchers to be able to plan conducting these suggested tests and parameters. Text reads: “Also, more direct parameters such as muscle oxygenation, maximal oxygen uptake, mitochondrial respiration, liver function test for determining the lactate clearance, and pulmonary function test which could improve the understanding of the observed issue were not conducted in this research, but are suggested to be evaluated in future studies.”
Thank you very much once again for such detailed suggestions on what could be done to gain a detailed view of the investigated issue, and we will try to include at least some of the proposed parameters/tests in future work.
Reviewer 2 Report
Comments and Suggestions for Authors
The objective of the evaluator paper is: This study aimed to investigate whether wrestlers of different competitive qualities (i.e., medalists vs. non-medallists) would differ in terms of specific fitness tests performance, cardiac, and metabolic responses after a demanding testing protocol.
The authors of the research report are requested to solve the following problems:
1) The objective declared in the abstract section is more precise than the one declared at the end of the introductory section (homogenize the objectives). The introductory section declares “creating a physically demanding testing protocol” (Line: 83-84), but does not demonstrate steps in this regard, (they have to specify that they are future actions). On the other hand, the objective declared in the first paragraph of the discussion section also varies significantly from that declared in the aforementioned sections; it is recommended to specify only the performance assessment indicators used, and not declare them in a general way.
2) It is necessary to specify whether the study sample belongs to the same weight division, since strength/speed indicators, among others, will vary depending on the athlete's body mass. The above is important, since the independent samples will have to have a certain degree of homogeneity.
3) The statement presented in lines 76-77 must be bibliographical references to support the clarified hypothesis.
4) Line 90 states a sample size of 29 Greco-Roman wrestlers; however, the normality test used was Kolmogorov‒Smirnov (Line: 176). The declared normality test is for samples greater than 50 subjects, for the case study the Shapiro-Wilk test must be applied. It is not known whether there is normality in the distribution of the data, as the authors of the paper did not declare it. However, the existence of normality is deduced by using parametric tests such as Student's t and ANOVA. However, it is unlikely that so much data from different tests will all be normal; Therefore, it is very likely that the authors will have to use non-parametric statisticians. (It is important that the authors of the research socialize the databases)
5) The discussion presents little debate with previously published works, the article has a limited bibliographic reference base.
6) Specify the strengths and limitations of the research at the end of the discussion section. Example: the sample size being small does not warrant generalizations of the results.
Comments on the Quality of English LanguageEvaluate the content with English language specialists
Author Response
Reviewer 2
The objective of the evaluator paper is: This study aimed to investigate whether wrestlers of different competitive qualities (i.e., medalists vs. non-medallists) would differ in terms of specific fitness tests performance, cardiac, and metabolic responses after a demanding testing protocol.
The authors of the research report are requested to solve the following problems:
- The objective declared in the abstract section is more precise than the one declared at the end of the introductory section (homogenize the objectives). The introductory section declares “creating a physically demanding testing protocol” (Line: 83-84), but does not demonstrate steps in this regard, (they have to specify that they are future actions). On the other hand, the objective declared in the first paragraph of the discussion section also varies significantly from that declared in the aforementioned sections; it is recommended to specify only the performance assessment indicators used, and not declare them in a general way.
RESPONSE: Thank you for noticing this. We unified the objectives throughout the manuscript now. Also, it is now explained that repeating the tests itself creates physically demanding testing. Text now reads: “Therefore, this study aimed to investigate whether wrestlers of different competitive qualities (i.e., medallists vs. non medallists) would differ in the performance, cardiac, and metabolic parameters of SWFT after repeating tests, which created a physically demanding testing protocol that mimics real-life match situations.”
- It is necessary to specify whether the study sample belongs to the same weight division, since strength/speed indicators, among others, will vary depending on the athlete's body mass. The above is important, since the independent samples will have to have a certain degree of homogeneity.
RESPONSE: Indeed, you are completely right. The weight of the wrestler is one of the crucial indices for this sport. Therefore, this issue has been addressed by allocating the dummies to each wrestler according to weight category as follows: the 55–67 kg category was tested with a 23 kg dummy, the 72–87 kg category was tested with a 25 kg dummy, and wrestlers weighing more than 90 kg were tested with a 30 kg dummy. As the SWFT was the test used for assessing all the study’s parameters, we believe that this procedure annulated the weight concerning performance. Please see the Methods section, Variables where SWFT is described. Also, we added the explanation now in that section, text reads: “Since the athlete's body mass plays a role in physical performance indicators, wrestlers performed the SWFT with specific weight of the dummy.”
- The statement presented in lines 76-77 must be bibliographical references to support the clarified hypothesis.
RESPONSE: The references which support the statement are now added.
- Line 90 states a sample size of 29 Greco-Roman wrestlers; however, the normality test used was Kolmogorov‒Smirnov (Line: 176). The declared normality test is for samples greater than 50 subjects, for the case study the Shapiro-Wilk test must be applied. It is not known whether there is normality in the distribution of the data, as the authors of the paper did not declare it. However, the existence of normality is deduced by using parametric tests such as Student's t and ANOVA. However, it is unlikely that so much data from different tests will all be normal; Therefore, it is very likely that the authors will have to use non-parametric statisticians. (It is important that the authors of the research socialize the databases)
RESPONSE: Thank you for this suggestion. We rerun the testing for normality with the Shapiro-Wilk W test and again got that all studied variables were normally distributed. We added this in the text to make it clearer, the text reads: “The Shapiro-Wilk W test was used to determine the normality of the distributions of all the variables included. The means and standard deviations were included as the descriptive statistics. Since all variables were normally distributed, the parametric tests were used to answer the research questions.”
- The discussion presents little debate with previously published works, the article has a limited bibliographic reference base.
RESPONSE: The discussion is now expanded and it includes more references to previously published similar research and a more detailed explanation for the results obtained. Accordingly, the bibliographic reference base is now expanded (from 22 to 33 references).
- Specify the strengths and limitations of the research at the end of the discussion section. Example: the sample size being small does not warrant generalizations of the results.
RESPONSE: The paragraph Limitations and strengths is now expanded, text reads: “The main limitation of this study is that wrestlers were tested at the end of the competition season, and testing during different points of the season could lead to different results. Thus, a similar testing procedure should be conducted at several main points during the competition season (e.g., beginning, mid-season, and end of the sea-son among the same wrestlers). Additionally, we included only male participants, so the results and conclusions from this study cannot be generalized to female participants. Moreover, the relatively small sample size does not warrant generalizations of the results which means that future similar studies should try to include a larger number of athletes. Also, more direct parameters such as muscle oxygenation, maxi-mal oxygen uptake, mitochondrial respiration, liver function test for determining the lactate clearance, and pulmonary function test which could improve the understand-ing of the observed issue were not conducted in this research, but are suggested to be evaluated in future studies.
The main strength of this study is that cardiac, metabolic, and performance variables were recorded at several time points during the testing procedure, which helps to track the response of the athletes to the given effort. In this way, it was possible to determine in detail which parameters are decisive for optimal performance and which are the most important parameters for competitive success. Also, the strength of this research is that competitive-level wrestlers were included, as it is always challenging for researchers to include elite athletes.”
Round 2
Reviewer 2 Report
Comments and Suggestions for Authors
The authors of the research have solved the problems indicated.